# Community's experience and perceptions of maternal health services across the continuum of care in Ethiopia: A qualitative study

**Gizachew Tadele Tiruneh**[1]*, **Meaza Demissie**[2], **Alemayehu Worku**[3], **Yemane Berhane**[2]

**1** The Last Ten Kilometers (L10K) Project, JSI Research & Training Institute, Inc., Addis Ababa, Ethiopia,
**2** Addis Continental Institute of Public Health, Addis Ababa, Ethiopia, **3** Addis Ababa University School of Public Health, Addis Ababa, Ethiopia

* gizt121@gmail.com

## Abstract

### Background

Continuum of care is an effective strategy to ensure that every woman receives a series of maternal health services continuously from early pregnancy to postpartum stages. The community perceptions regarding the use of maternal services across the continuum of care are essential for utilization of care in low-income settings but information in that regard is scanty. This study explored the community perceptions on the continuum of care for maternal health services in Ethiopia.

### Methods

This study employed a phenomenological qualitative research approach. Four focus group discussions involving 26 participants and eight in-depth interviews were conducted with women who recently delivered, community health workers, and community leaders that were purposively selected for the study in West Gojjam zone, Amhara region. All the interviews and discussions were audio-taped; the records were transcribed verbatim. Data were coded and analyzed thematically using ATLAS.ti software.

### Results

We identified three primary themes: practice of maternal health services; factors influencing the decision to use maternal health services; and reasons for discontinuation across the continuum of maternal health services. The study showed that women faced multiple challenges to continuously uptake maternal health services. Late antenatal care booking was the main reasons for discontinuation of maternal health services across the continuum at the antepartum stage. Women's negative experiences during care including poor quality of care, incompetent and unfriendly health providers, disrespectful care, high opportunity costs, difficulties in getting transportation, and timely referrals at healthcare facilities, particularly at health centers affect utilization of maternal health services across the continuum of

**Data Availability Statement:** All relevant data are within the paper and its Supporting information files.

**Funding:** JSI Research & Training Institute, Inc. has provided us support in the form of salaries for author [GT]. However, any of the funders did not have role in study design, data collection and analysis, decision to publish, or preparation of the manuscript.

**Competing interests:** The authors declare that they have no competing interests. One of the authors have been working for JSI Research & Training Institute, Inc., a commercial company. We declared that this commercial affiliation does not alter our adherence to PLOS ONE policies on sharing data and materials.

**Abbreviations:** ANC, antenatal care; CoC, continuum of care; FGD, focus group discussion; HEP, Health Extension Program; HEW, Health Extension Worker; IDI, in-depth interview; L10K, Last Ten Kilometers Project; LMP, last menstrual period; PNC, postnatal care; WDA, Women Development Army.

care. In addition to the reverberation effect of the intrapartum care factors, the major reasons mentioned for discontinuation at the postpartum stage were lack of awareness about postnatal care and service delivery modality where women are not scheduled for postpartum consultations.

## Conclusion

This study showed that rural mothers still face multiple challenges to utilize maternal health services as recommended by the national guidelines. Negative experiences women encountered in health facilities, community perceptions about postnatal care services as well as challenges related to service access and opportunity costs remained fundamental to be reasons for discontinuation across the continuum pathways.

## Background

Maternal and newborn health is a major public health problem in low-income countries. Sub-Saharan Africa (SSA) is the only region with unacceptably high maternal mortality ratio, accounted for about two-thirds of global maternal deaths in 2017 [1]. Though Ethiopia achieved a substantial reduction in maternal death of roughly 61% between 2000–2017 [1], its maternal mortalities are still among the highest in the world [1–3] and persistently high neonatal mortality since the last two decades [4–6]. The utilization of maternal and newborn healthcare services across the continuum remains low in SSA [7–9]. The uptake of services drastically declined from antenatal to the postnatal period, along with the CoC [10] where coverage is lowest during childbirth and postnatal period, and services are often fragmented and weakly implemented, limiting continuity of care [11].

Communities' perceptions about health programs and health services greatly affect the uptake of recommended maternal health services in low-income settings [12–14]. The Andersen and Newman behavioral model for health service utilization provides a framework to describe factors that influence individual decisions to use health care services. Accordingly, the utilization of maternal health across the continuum of care (CoC) is a function of characteristics of the health system, individual characteristics, and interaction of individual and societal determinants [12].

The main health system factors that have contributed to the failure to deliver effective maternal health interventions are shortage of trained staff, poor quality of care, delayed use of services and low demand for care, erratic supply of essential drugs and supplies, and institutional segmentation of the health system, and inadequate operational management [11, 15]. The quality of antepartum and intrapartum care influences women's intention towards having follow-up visits and practicing key postpartum maternal health behaviors [16].

Individual determinants of health services utilization are a function of their predisposition to use of services, factors which enable or impede use, and their need for care [12]. Different literature shows maternal care utilization is influenced by various perceived barriers including socio-cultural, economic and physical accessibility, and perceived benefit/need factors [13, 17]. Recent studies in Ethiopia show, continued use of services varied significantly across wealth, antenatal care (ANC) visits in the first trimester, quality of ANC, and mode of delivery across the CoC [18, 19]. The need for a continuum of maternal health care can also be affected by the intrapartum and postpartum complications or duration and severity of disabilities

encountered, individual past experiences with pregnancy, childbirth and health services, self-perceived health status, and perceived quality of care [17].

Evidence shows that individual supply-side factors alone do not fully explain the variability in women's ability to use maternal health care [20]. Social norms—informal rules of behavior that dictate what is acceptable within a given social context—influence health-related choices and behaviors of people [21, 22]. In low-income settings, religious beliefs, social norms about pregnancy and childbirth care, values and opinions of other family and community members, and the need to seek permission from husbands and other family members significantly influence women's decision to seek care [23–26]. As a result of gendered social norms, in some societies, women's autonomy and decision-making power to access to and control over resources and negotiation power with her partner and family are limited that restricts their ability to make decisions with regards to maternal health and health care [27–30].

In low-income countries, community engagement strategies and community health worker programs are used to improve access and utilization of health care services through bridging families, communities, and health facilities [31]. Ethiopia has employed different community engagement approaches for decades. The Health Extension Program (HEP), launched in 2004, is an institutionalized community health system that engages communities to identify their priorities and solve their health problems [32]. Evidence shows that HEP has contributed to improving community access to health services and it has been associated with increases in the uptake of maternal health services [33–35]. Nonetheless, the utilization of maternal healthcare services across the continuum of maternal health care remains low [8]. Further analysis of the Ethiopian Demographic and Health Survey 2016 showed that 45%, 47%, and 87% of women who booked for antenatal care visits dropped from a recommended number of ANC, institutional delivery, and postnatal care (PNC) visits, respectively [36].

Undue considerations of the role of community perceptions and negative experiences in planning and implementation of maternal health services contribute to poor maternal health-care uptake in rural communities [24, 29, 37]. This is mainly due to lack of adequate evidence and comprehensive understanding of the communities' perceptions and experiences of maternal health care. In Ethiopia, although there are few studies on factors associated with the adherence to the continuum of maternal health care, community perceptions regarding the use of maternal services across the continuum of care are not well documented. As such, this study explored the community perceptions on the continuum of care for maternal health services in Ethiopia for a deeper understanding of women's experiences and challenges regarding the use of skilled care at antepartum, intrapartum, and postpartum stages. Therefore, understanding community perceptions of health and experiences of care can serve as an evidence base for community-based maternal health initiatives.

## Methods

### Settings

The Ethiopian health system is structured into primary level care, secondary level care, and tertiary level care. The primary care structure found in a *woreda* (i.e., district) health system comprises a primary hospital, 4–5 health centers, and 20–25 health posts for the delivery of basic curative, preventive and promotive community and outreach services with a seamless continuum.

Ethiopia has employed different community engagement approaches for decades through the use of voluntary community health workers with various names and scopes of practice including community health agents, community-based reproductive health workers, community health promoters or volunteers, and traditional birth attendants. Since 2004, the country

has expanded community health services through the expansion of the HEP and actively engaging community volunteers to reach most communities and households [32]. In 2011, the community engagement has restructured and introduced the Women Development Army (WDA) strategy to further strengthen the HEP, and participation of individuals, families, and communities. Under the WDA strategy, women are organized and mobilized in groups to share actionable messages and influence each other to adopt and practice healthy behaviors [38].

Women development groups support Health Extension Workers (HEWs) in promoting key messages related to skilled maternal health care through social events such as coffee ceremonies, using peers during marketing, and other community events. They identify pregnant women and birth in their communities and link them to HEWs for early ANC and PNC care.

The study was conducted in five rural woredas namely Burie Zuria, Dembecha, Jabi Tehnan, Quarit, and Womberima in West Gojjam Zone of Amhara region. The zone and districts were purposely selected based on the feasibility of establishing sampling frame as this zone has only five The Last Ten Kilometers (L10K) Project woredas while other L10K zones have 10 woredas, and availability of the established relationship with the zone and woredas which allowed practical feasibility to do the data collection. This helped the researchers to minimize difficulties and expenses involved in the planning and conduct of data collection.

## Design and population

We utilized a phenomenological qualitative research approach. Based on the Andersen and Newman Framework for health services utilization model [12], we hypothesized that the continuous uptake of maternal care is influenced by the women's and communities' perceptions and previous experiences of care, women's exposure to the health system at any stage of the antepartum, intrapartum, and postpartum stages, and health service and program-related factors. Accordingly, guided by this model, mothers' and community's experiences and perspectives regarding maternal health services were captured through in-depth interviews (IDI) and focus group discussions (FGDs) to answer the research questions in detail that is to gain individual and group/community perceptions and experiences regarding maternal health services.

Three different groups of participants were selected to gain as much insight and understanding as possible about maternal health care services from the community's perspectives. It is composed of women 15–49 years of age who have given birth in the last year before the date of data collection, community elders and community and religious leaders (all were males and hereafter referred to as community and religious leaders), and community volunteers (i.e. WDAs).

## Sampling methods

Maximum variation sampling schemes were used to yield a wider perspective from various groups of stakeholders. We subdivided the *woredas* (i.e. districts) into better performing and low performing strata in terms of maternal health service utilization based on routine administrative data obtained from the zonal health department. Accordingly, two of the study woredas were classified as better-performing, and the other three as low-performing woredas. In each woreda, *kebeles*, the smallest administrative unit, were selected based on the feasibility of convening FGD participants and availability of HEWs in the Kebele. Lastly, from each kebele, study participants were recruited with the assistance of HEWs based on pre-set selection criteria that include having lived experiences of maternal health services, being recognized as influential and motivator of maternal health service uptake in the community, and being a community volunteer or WDA.

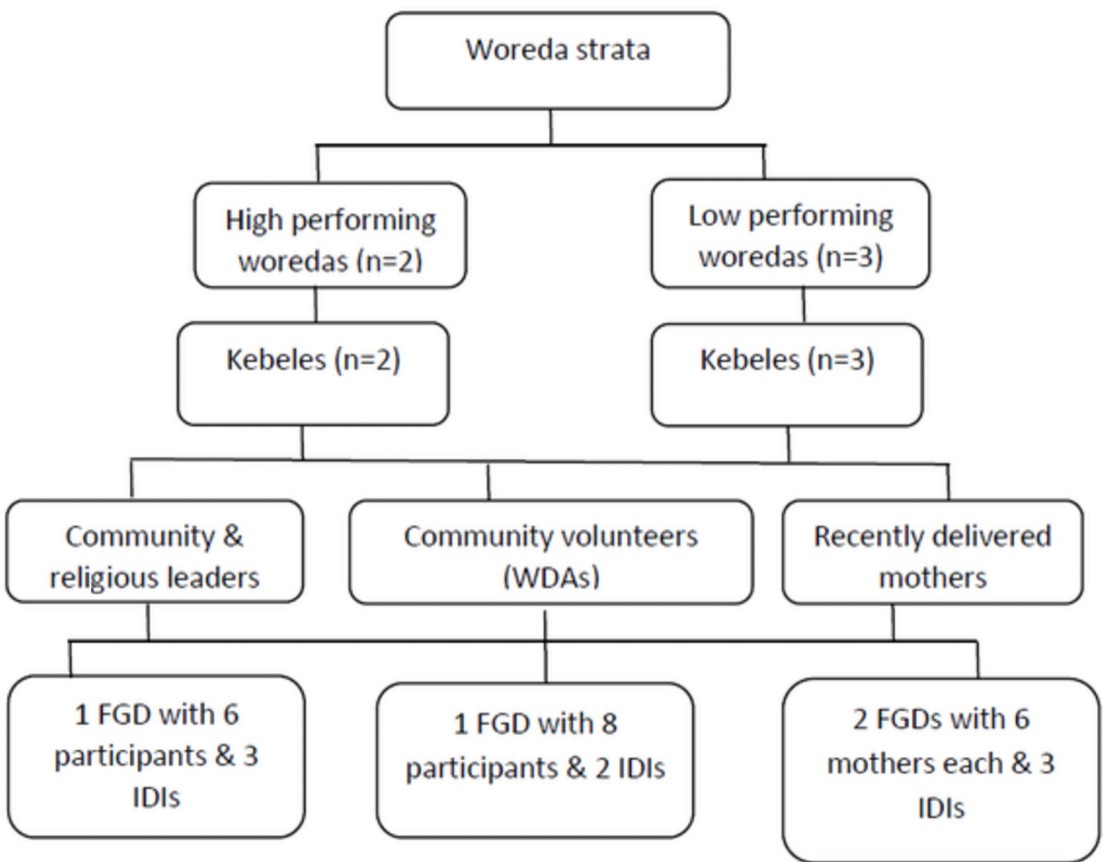

**Fig 1. Sampling frame for the qualitative study.** The principal investigator and research assistants collected and analyzed informational redundancy, data saturation, after conducting three FGDs and six IDIs [39].

Fig 1 presents the detailed sampling frame and respondents' categories.

## Data collection

Data were collected in October-November 2019. Two research assistants who had experience in qualitative research along with the principal investigator collected the data.

Interview and FGD guides were prepared with open-ended questions with probing questions (S1 and S2 Appendices) were used to capture the required information from study participants. The interview and discussion guides were prepared in English and translated to Amharic, the local language. The main topics of discussion included community perceptions about maternal health programs and health providers; the practice and experiences of antepartum care, facility delivery, and postpartum services; and reasons for discontinuation across the continuum of maternal health care. Then, follow-up questions were asked to help to explore their perceptions and experiences of receiving maternal health services in detail. Interviews and group discussions were conducted at a convenient place in the community.

Focus group discussions were used to explore information about the social context and discussed the differences among participants. The size of the FGD ranged from 6–8 participants to elicit group-level perceptions by facilitating active interaction. The FGD sessions lasted between 49 and 99 minutes (discussions with WDAs lasted longer which might be due to their

lived experience as a mother as well as their engagement in promoting maternal health messages as a community volunteer).

Individual interviews were conducted with recently delivered mothers, WDAs, and community and religious leaders to investigate personal perspectives; some private issues, for instance, delivery experience; issues that were raised during FGD that needs further investigation; and to identify opportunities for improving maternal health services. Interview sessions lasted between 21 and 43 minutes.

The principal investigator ensured the quality of the data by conducting regular reflective discussions with the research assistants. These discussions were held between the interviews or discussion sessions to discuss key findings, refine the FGD and IDI guides, and identify strategies that continually enhance the line of inquiry following the tradition of emergent design in qualitative research. Throughout the data collection, the study participants were probed to elaborate on or clarify what they have said during the interviews or group discussions to confirm the accuracy of the information captured and the meanings that the participants intended to ascribe to.

Throughout the data collection and archive, we ensured confidentiality of the data. Individual personal identifiers were not collected and authors did not have access to information that could identify individual participants during or after data collection.

## Data analysis

All interviews and discussions were audio-taped with the consent of the study participants and the records were transcribed verbatim by the principal investigator and the research assistants. The principal investigator listened to the audio records and read through the transcripts several times to have an overall sense of the data and to organize the transcripts.

The transcript texts were exported to ATLAS.ti software for analysis. A transcript analysis approach was employed which involved familiarization, coding and categorizing data, identifying themes and interpretation stages. Both deductive and inductive coding approach was applied. Guided by the conceptual framework and interview and discussion guides, predefined initial codes were developed (open coding) before data collection. Then, each code was further analyzed and disaggregated into categories and sub-themes (deductive axial coding). Iteratively, through reading the data, all data were subsequently classified into one of the codes. Additional codes were added while reading the data, categories, and sub-categories that had not been previously identified (inductive approach). Data were triangulated from responses obtained from IDIs and FGDs to compare them with responses from the different community groups.

The categories and the concepts that emerged from interviews and discussions were verified by consistently linking the emerging categories with the data received from the other groups of informants to improve the trustworthiness of the qualitative data analysis. Quotes were used to enhance credibility and substantiate the narrative with participants' own words.

Reports of quotations for selected codes were generated in ATLAS.ti software. Themes and patterns of interrelationships between the themes were identified and reported.

## Ethics approval and consent to participate

Ethical clearance was obtained from the Research and Community Service Office of the University of Gondar (reference number V/P/RCS/05/2505/2019; dated on 25 August 2019). The objectives, methodology, purpose of the study, and the benefits and risks of the study were explained to all study participants. Before data collection, participants were also informed of their right to voluntarily participate in the study. Verbal consent was sought and documented

before conducting any interviews and discussions. Because the majority of the respondents were not expected to be able to read or write; written consent was not sought. If the respondent agreed to be interviewed after listening to the consent statement, the interviewer marked the consent form as consent given below the consent statement and signed below that. The interviewer continued with the interview only after receiving and documenting consent. The survey protocol submitted to the ethical review committee included the study questionnaire with the statement that described the consent-obtaining procedure. The primary investigator oriented data collectors on ethical and methodological issues and supervisors followed them throughout the data collection period to monitor any ethical breach.

To ensure the privacy of study participants, respondents were interviewed in a conducive environment; and confidentiality of the data were guaranteed by preserving the anonymity of the study participants. Individual personal identifiers were not collected, to ensure the anonymity of data and the researcher would keep the information obtained from the research participant in private or will withhold information from others, to respect the confidentiality and privacy of study participants. Identifying information (names and addresses) was not included in the data collection instrument.

## Results

### Study participants

A total of four FGDs involving 26 participants and 8 IDIs were conducted: 3 IDIs with delivered mothers; 2 IDIs with community health volunteers/ WDAs; and 3 IDIS with the community and religious leaders.

Participants in FGDs included 12 recently delivered mothers, 6 WDAs, and 8 community and religious leaders. The mean age of mothers was 27 years raged from 20–34 years. Most mothers were illiterate but few completed 4–10 grades. While the mean age of WDAs was 47 years ranged from 42–55 years. They serve for a median of 6 years as a community volunteer. On the other hand, the mean age of community and religious leaders were 52 years ranged from 41–72 years.

### Key findings

We identified three primary themes including views on and practice of maternal health services; factors influencing the decision to use maternal health services across the continuum of care; and reasons for discontinuation across the continuum of care for maternal health services. Under each thematic area, several categories were identified (Table 1).

### Views on and practice of maternal health services across the continuum

**Perceived need and uptake of maternal health services across the continuum of care.** According to informants, nowadays, most women utilize ANC, facility delivery, infant vaccination, and postpartum family planning services. They mentioned that the situation has now changed due to the provision of health education by health workers, the establishment of health centers in their vicinity, the presence of HEWs, and other similar efforts.

"*Nowadays, no pregnant women stay home without having ANC; all are aware of the importance [of ANC consultations].*"

**(recently delivered mother, FGD).**

**Table 1. Major themes and categories identified.**

| Themes | Categories |
|---|---|
| Views on and practice of maternal health services | • Uptake of maternal health services across the continuum<br>• Timing and frequency of visit<br>• Experiences of care |
| Factors influencing decision to use maternal health services across the continuum | • Health promotion activities<br>• Fear of obstetric danger signs and previous experiences of complications<br>• Community support |
| Reasons for discontinuation across the continuum of maternal health services | • Late ANC care-seeking<br>• Perception and fear of overgrowth of the fetus<br>• Perceived poor quality of care<br>• Disrespectful care<br>• Lack of trust in the competence of providers<br>• Service inaccessibility<br>• High opportunity cost<br>• Lack of understanding about PNC services<br>• Lack of PNC service or schedule at health facilities |

Most participants mentioned the benefit of receiving skilled care during pregnancy and childbirth. They acknowledged the risk of complications and the risk of death associated with pregnancy and childbirth. However, there is no formal PNC service for both the mother and the newborn. After delivery, women seek care only if they are sick, otherwise, they would only seek vaccination and family planning services after 45 days postpartum. A recently delivered woman affirmed this, *"No, we just stay home, we do not go if we are not sick."* (**recently delivered mother, FGD**).

**Timing and frequency of ANC visit.** According to the respondents, most women started booking ANC at or about 3 months or later when they would be certain about the continuity of their pregnancy or when they felt ill-health. One of the community leaders said, "*In about third month of their pregnancy, the majority of pregnant women . . . would go for pregnancy screening and start pregnancy follow-up.*" (**community leader, FGD**) Culturally, women did not believe that one can be certain about the continuity of their pregnancy before three months. It is believed that the pregnancy might end in miscarriage if revealed in the earlier periods. Thus, there is a taboo to disclosing pregnancy.

"*I think they do not believe their pregnancy could continue and some of them do not want to disclose their pregnancy before three months for fear of miscarriage.*"

**(WDA, IDI)**

Pregnancy with unknown last menstrual period (LMP) could be the reason for delayed booking for ANC, "*Probably, they may not know their LMP; they started booking after they received a pregnancy test at the health center. That is why they elapsed one or two months to start ANC booking.*" (**WDA, IDI**) Moreover, according to the accounts of respondents, family planning methods women used might cause an absence of menstrual period and this might mislead women, becoming a reason for them not to go and test for pregnancy and thus the reason for not start their ANC early.

Regarding the frequency of visits, usually, women receive about 4 ANC visits. Study participants mentioned that some mothers might not receive the recommended number of ANC consultations due to late booking and perceived fear of overgrowth of the fetus. A community volunteer said, *"If she feels sick, she would visit the health center before her scheduled visit and thus may end up visiting the facility four or five times. However, if she is not sick, she would visit four times."* **(WDA, IDI)**.

**Experiences of care.** Most study participants appreciated the services they received from health professionals at hospitals, health centers, and health posts. They mentioned that *"The health center and health professionals are compliant and willing to serve the community."* **(community leader, IDI)** Besides, respondents witnessed that they have trust in facility care. A community volunteer said, *"We rely on health facility care. We feel that no death occurs there."* **(WDA, IDI)** However, some respondents mentioned that they experienced negative interactions with providers, delays in care and service provision, and delay in referral to the next level of care, particularly during intrapartum care at the health center level.

*Negative interactions with providers.* Some study participants mentioned that they have experienced negative interactions with providers including abusive care from providers, guards, and ambulance drivers and discrimination based on their socioeconomic status. Community leaders who were participants of FGD said that some facility staff behavior and approach towards clients is not good at all. The mother expressed her experience, *"while I was vomiting, she [the provider] insulted and pushed me to come down [from the coach]."* **(recently delivered mother, FGD)**.

*Delays in service provision and referral.* Despite the high perceived need for maternal services across the continuum of care, respondents mentioned that mothers also experienced delays in getting care at the facilities and referral to the next level of care. Discussions with the study participants also indicated that the delay in care provision resulted in poor interaction with providers. One of the participants expressed her concerns, regarding such experiences she observed, as follows,

> *". . . the health professionals usually play cards while a pregnant woman is screaming and suffering from labor pain instead of monitoring her. When we ask them to refer her to other health facilities if they cannot do anything to help her, they become belligerent and tell us to leave and take her away if we want to. No one follows her to see if she is bleeding or not or had lost fluid [amniotic fluid] early because they do not monitor her status regularly."*

> **(WDA, FGD)**

Respondents also experienced poor quality of the delivery services at the health center. *"The health center does not give us sufficient attention and follow-up."* **(recently delivered mother, FGD)**. Accordingly, women preferred to go to the hospital or deliver at home.

> *". . .. Many women do not want to give birth in this health center and prefer to be referred to a nearby hospital."*

> **(recently delivered mother, FGD)**

The FGD participants claimed that they had experienced delays in the referral process while being referred from the first-level health facilities to higher-level facilities. Respondents claimed that the health center staff did not initiate timely referrals.

## Factors influencing decision to use maternal health services across the continuum

Based on participants' opinions, health promotion activities, previous experiences, and community support facilitate the use of skilled maternal health care across the continuum.

**Health promotion activities.**   The health services provided by community health workers and health professionals influenced women to utilize maternal health services.

> "*Previously, before the introduction of the ANC service, pregnant women used to give birth in their homes and made no ANC check-up; some died and others survived by chance in the process. However, after the introduction of ANC service and the education we got from the HEWs, we can identify pregnant women early . . . and convenience them to go and start pregnancy follow-up.*"

**(WDA, FGD)**

**Fear of obstetric danger signs and previous experiences of complications.**   Previous experience of obstetric problems and complications and fear of happening again influenced women to utilize maternal health services continuously.

> "*My wife did not like going to the health center for follow-up care. In her first pregnancy, she almost died because the fetus appeared upside-down [breech presentation] during labor; the fetus died. We learned from that experience and in the second pregnancy she attended ANC, the fetus presented same way as before, breech, the health facility referred her to hospital on time and she gave birth safely.*"

**(community leader, FGD)**

Another FGD participant supported the above comment as follows, "*My wife had experienced severe bleeding after delivery during her previous pregnancies. In her last pregnancy, I took her to the facility.*" **(community leader, FGD)**.

**Community support.**   Respondents described that communities, particularly their neighbors, provided support to the women in managing household chores during pregnancy and childbirth. They take care of the children, manage household assets and activities such as cultural ceremonies. The community played a facilitative role in laboring women by calling for ambulance services and informing HEWs. Furthermore, they also indicated that the communities transported the women to an ambulance access point or health facility and again back to their homes, particularly in circumstances when there was no access to ambulance services. "*If the village is far, the community will take her [the pregnant woman] on a locally made carrier.*" **(WDA, FGD)**.

## Reasons for discontinuation across the continuum

Late ANC booking and fear of the overgrowth of the baby were mentioned as reasons for discontinuation of maternal health services at the antepartum stage. On the one hand, multiple reasons were mentioned for discontinuation at the intrapartum stage that includes: poor quality care, incompetency of providers, disrespectful care, inaccessibility to service, and high opportunity costs. On the other hand, in addition to the reverberation effect of the intrapartum care factors, the major reasons for discontinuation at the postpartum stage were lack of awareness about PNC care and lack of PNC service or inconvenience of the schedule at health facilities.

**Late ANC booking.** Those mothers who booked ANC at a later trimester of their pregnancy might get less than four ANC visits and may not complete the recommended antepartum services. "*. . .for instance, a woman in our locality did not realize she was pregnant until the fourth or fifth month of her pregnancy. This delayed her ANC follow-up and a few months later she was in labor.*" *(***WDA, FGD***)* In line with this, study participants also mentioned that women's physical strength in the last trimester of their pregnancy which makes it difficult to travel on foot long distances to seek care is one of the reasons for discontinuation of antepartum care.

**Perception and fear of overgrowth of the fetus.** Women's perception and fear of overgrowth of the fetus when women received frequent antepartum care, influenced some women to discontinue their antepartum follow-up. Participants affirmed that some women believed that frequent ANC check-ups would result in overgrowth of the fetus that in turn would result in prolonged labor as the following quote demonstrates. "*There are some women who stopped their ANC follow-up for fear of fetal overgrowth and later prolonged labor as a result.*" (**recently delivered mother, FGD**).

**Perceived poor quality of care.** There is a reverberation effect of perceived poor service provision on the uptake of maternal health services across the continuum. The experience and perception of receiving poor quality service at health facilities during the antepartum visits for some and for others who received poor intrapartum care deterred women from continuing to use skilled delivery from the health facility. The study participants mentioned that women do not receive appropriate support from health facilities and had encountered problems during labor and delivery. As a result, they preferred to give birth at home rather than at health facilities.

"*. . .most would ask why they should go to deliver at the health center, for there is little the health center would do to help them. Most make their ANC follow-ups but when it's time to give birth, they try to seek assistance from TBAs since they are not satisfied with the service at the health center.*"

**(recently delivered mother, FGD)**

Delays in care and service provision particularly during labor and delivery in health facilities have created disappointments for women and the community. It was indicated that participants experienced delays in receiving maternal health care services from health facilities due to the unavailability as well as negligence of health workers.

"*. . .during her labor, there was a delay and when we asked him if he was waiting until she dies, he replied with arrogance telling us to wait, anyway. The mother in labor was very uncomfortable and told us to stop saying anything. Finally, she was able to deliver with the help of St. Mary and she made it back home.*"

**(WDA, FGD)**

**Disrespectful care.** Some women avoided accessing maternal health services due to abusive behavior, discriminatory attitude of health professionals, non-compassionate care as well as lack of autonomy. Community leaders who were participants of FGD said that some facility staff behavior and approach towards clients is not good at all. They said that the staff are arrogant and commented saying this should be corrected.

"*One day, I took my wife who was in labor to a health facility. The health provider on-duty shouted at me that "she is not in her term and labor" and pushed me out. She stayed in the*

*delivery room and I was waiting outside far away from the delivery room, after a while he called me to come into the delivery room since my wife had to deliver.*"

**(community leader, FGD)**

Lack of supportive and compassionate care during labor and delivery was a major problem mentioned by the mothers as a reason for not attending skilled care. An FGD participant, a woman who recently delivered said, *"Most women question what would the health professionals do for them during their labor other than sitting outside and giving them no attention. Some women remain* home during their labor for these reasons." **(recently delivered mother, FGD)**.

Study participant women mentioned that *"The health professionals care better for the people coming from urban areas and not for us who come from the rural areas to whom they give poor consideration."* **(recently delivered mother, FGD).** They added that **"***They give more attention when they see educated people and not to us, the rural dwellers. Some pregnant women remain home during their labor because of this."* **(recently delivered mother, FGD)**.

Delivery positions, exposing their genitalia, and vaginal examination are disliked by most women. "One of the main *reasons for the dislike of institutional delivery is the position and vaginal exam. The vaginal exam is a very painful and disgusting experience."* **(recently delivered mother, FGD)** They mentioned these are the main reasons that frighten others from coming to the facility to deliver. A community leader expressed the community opinions by saying, "*There are women who give birth at home without disclosing their pregnancy. It is all about privacy. The issue is not distance or cost."* **(community leader, IDI)** The vaginal examination and lithotomy position affect other mothers who had no previous experience of the examination. This is how it was described by a woman who recently delivered. "*We fear the vaginal examination. Mothers who heard of examination by health workers inserting their hands get afraid and decide not to attend facility delivery."* **(recently delivered mother, FGD)**.

**Lack of trust in the competence of providers.** This study showed that mothers are dissatisfied with the intrapartum care provided at the health center level. Accordingly, they were inclined to give birth at home.

"*Women may go to the health center to give birth because it is mandatory, but most would ask why they go to the health facility as there is very little the health center does to help them. Therefore, most make their ANC follow-ups but at the time of delivery they opt for TBAs.*"

**(recently delivered mother, FGD)**

Women specifically mentioned skill deficiency regarding misdiagnosis and mismanagement of obstetric complications as well as identifying twin pregnancy and fetal presentation. The competence of care providers regarding obstetric care was found to be a major reason for women to discontinue maternity care and instead opted to give birth at home. The lack of confidence in skilled birth attendants was expressed by women as follows:

"*At the health center, I was told my pregnancy was fine the whole time, they did not notice the presence of twins. . . . but I was aware and knew my status because of my previous experience. After I arrived at the hospital, being referred, they told me that I was having twins.*"

**(recently delivered mother, FGD)**

Some women perceived that the midwifery skills for obstetric care at health centers were limited compared to those at hospitals. For this reason, they tended not to utilize maternal health services at health centers.

"*Some women say the health center is not up to the level to help them deliver and as a result, the women visit other health facilities.*"

**(recently delivered mother, FGD)**

**Service inaccessibility.**   Distance to the health facility, poor road conditions, and inadequate transportation were commonly cited as reasons for not utilizing maternal health services continuously. Access to ambulance service, particularly during the night time, was very challenging. "*Ambulance service is almost nil. We have to carry the laboring women to the bus stop or facility.*" **(community leader, FGD).** Mothers who participated in the FGD added that "*. . . On one occasion, we could not find an ambulance and finally, we tried to take her ourselves to health facility but she delivered on the way.*" **(community leader, FGD).**

Distance to the health facility and poor road conditions particularly during the rainy seasons were major barriers to seek skilled care for some women in certain distant communities. Community and religious leaders described the situation as follows,

"*The river in between the kebele and the health center is the biggest challenge for us. It is impossible to cross the river from June to September and during this time no one has access to service provided at the health centers. Previously there was a temporary bridge, but it was damaged by seasonal flooding due to heavy rains. Thus, pregnant women suffer a lot. For example, in the last rainy season (June to September 2019) four laboring women died [in this village].*"

**(community leader, FGD)**

**High opportunity cost.**   Though participants acknowledged the free maternity care services, there are high opportunity costs involved for maternity service utilization that hinder mothers from seeking intrapartum care. Opportunity costs including transport costs for returning home, food costs for accompanies, and medical costs. These are some of the barriers that hinder mothers from seeking intrapartum care. Focus group discussions among community and religious leaders revealed that if women become critically ill and feel they are going to die; they would not take them to the facility for fear of the cost of transport back home.

"*The other problem is the ambulance which does not provide round trip service and we have no other means of transportation to get back home. This is a big challenge for us. Especially some community members do not take critically ill women (pregnant or laboring) to the health facility, because if she dies at the health center or hospital returning the dead body will be too expensive, about 3,000–4,000 birr—[equivalent to USD 102–136].*"

**(community leader, FGD)**

Ambulance service is not easily accessible for facility delivery. Moreover, there is no transportation service provided to return home which is a big problem for the community. Some contract transport by paying up to 100 birr—[~ USD 3.4], "*. . . they do not return us home, and we spend 100 birr for Bajaj transportation to return home.*" **(recently delivered mother, FGD).**

Others traveled long distances carrying the mother and newborn; relatives/neighbors help. Accordingly, the refreshment fee (round the trip) for people carrying the woman is another cost involved for facility birth. In case of a referral from a health center to a hospital, the ambulance allows two people to accompany the woman. Other relatives/neighbors use public transport to accompany her.

"*Up to two people will be allowed into the ambulance while transporting the woman to the health center during her labor. The rest who accompany her will have to use their means of transport by covering their cost. The woman covers her own cost for transportation when she returns home after delivery.*"

**(WDA, FGD)**

All discussants and informants acknowledged the medical cost or cost of maternity services being free. But for prescriptions, they would pay out of their pocket, and reimbursement for their health insurance is difficult.

"*Regarding reimbursement for an outside health facility prescription fee, they responded that reimbursement is done at the woreda office, not at the health center and it is too far from the health center which is too difficult for us*".

**(community leader, FGD)**

**Lack of understanding about PNC services.** Most participants described that they are not receiving follow-up service after delivery. "*Unless it is for the baby's vaccination we do not make PNC for us. We go to the health center for the baby's vaccination 45 days after birth.*" **(recently delivered mother, FGD)**.

Some study participants argued that postnatal care is required only if they experience illness or complications. Participant women who recently delivered said, *"It's because we have no illnesses. Why would we go if we do not have any pain*!?" **(recently delivered mother, FGD)**. Community volunteers also affirmed this, "*Mothers do not believe postpartum is necessary."* **(WDA, IDI)**.

Most participants do not know about postpartum services. They described postnatal care as just being for immunization service or family planning services which they received at or after 45 days of postpartum. "*We went on the 45$^{th}$ day as per our appointment. For what purpose do we need to go before that*?" **(recently delivered mother, FGD)**.

**Lack of PNC service or schedule.** Most women mentioned that they were not informed about postpartum visits to the facility. Mothers in the FGD said, **"***We have not been told by the health center that there is PNC service for us, only vaccination for the baby." **(recently delivered mother, FGD)** Health providers advised them to return for child vaccination 45 days after delivery. "*They told us to come back on the 45$^{th}$ day for vaccination and family planning. . . . we went on the date of appointment."* **(recently delivered mother, FGD)** "*After 45 days, however, they visit the health facility to get family planning services for them and vaccination services for their babies."* **(WDA, IDI)**.

## Discussion

This study explores community's perceptions and experiences about the continuum of care for maternal health services; factors influencing the decision to use maternal health services across the continuum; and reasons for discontinuation across the continuum of maternal health services in rural Ethiopia. The findings show that though communities have positive attitudes and good practices regarding continuous use of skilled antenatal and delivery care, most women dropped receiving care at the postpartum stage. Besides, some respondents experienced negative interactions with providers, delays in care and service provision, and delay in referral to the next level of care, particularly during intrapartum care at the health center level. Our results also demonstrate that communities' perceive fear of obstetric danger signs and previous experiences of complications, health promotion activities, and community support were

influencing the decision to use maternal health services across the continuum. Regarding reasons for discontinuation, communities perceive a wide range of factors that contribute to the discontinuation across the continuum of time dimensions in Ethiopia. Late ANC booking, fear of overgrowth of the baby and thereby fear of prolonged labor were the main reasons for the discontinuation of maternal health services across the life course continuum at the antepartum stage. Perceived reasons identified, in this study, as factors for women's fragmented use of maternity care at the intrapartum stage and giving birth at home include perceived poor quality of care, disrespectful childbirth care, perceived limited obstetric competence of providers, lack of healthcare access, and high opportunity costs. In addition to the reverberation effect of intrapartum care factors, the major perceived reasons mentioned for discontinuation at the postpartum stage were lack of awareness about PNC and service delivery modality where women were not scheduled for postpartum consultations.

Most communities in the study area had positive attitudes as well as practices of antenatal care and facility delivery which is in line with recent national demographic and health survey findings in Ethiopia where about three-fourths of women had at least one ANC visit with a skilled provider and over the last decade, i.e., between 2011 and 2019, the institutional delivery levels showed a five-fold increase [40]. This study also identifies health promotion activities and community support were positively influencing the decision to use maternal health services across the continuum. This indicates the importance of strengthening the implementation of community-based health promotion and education interventions, engaging communities, and for sustained behavior to improve utilization of maternal health services.

Late ANC booking for consultation is an important factor for discontinuation of maternal health services across the continuum. This is in line with previous studies documenting women who book their ANC care when in their later trimester, would receive fewer ANC visits that result in discontinuation of subsequent maternal health services across the continuum pathway [7, 19, 41].

Perceived poor quality of maternity care has an echo effect on the uptake of maternal health services across the continuum. The experience of receiving poor quality service during the antepartum visits or poor intrapartum care deterred women from continuing to use skilled delivery from the health facility. Due to the poor quality of health center level obstetric care, some communities prefer traveling directly to hospitals or to be referred to one. Due to perceived delay of care and referral, clients usually request timely care or timely referral. Referral requests made by clients result in poor interaction with the health center staff which is linked with the abandonment of care and perceived poor competence of providers. This also shows that the community sometimes considers the quality of care more than the opportunity cost which is incurred by traveling long distances to hospitals. However, as they indicated, when a referral is initiated late and the family feels that the woman is too tired, they would decline the referral and travel back home fearing the opportunity costs that would be incurred in transporting (what they believe would be) a dead body. Previous studies also documented poor quality of primary care limits the utilization of continuous maternal health services [11, 13, 15]. Therefore, service providers and program managers need to closely monitor the maternity services delivered at the primary level of care to improve the quality of care and client-provider interaction.

The findings of this study identified disrespectful maternity care as a major barrier to women's continuous use of primary health care facilities, especially for childbirth. Disrespect and abuse during childbirth, discrimination based on socio-economic status, and poor client-provider interaction, influence the use of maternal health services across the continuum. Unduly exposure of women's reproductive organs, lack of privacy, and repeated and painful vaginal examination practices were also found to be other barriers for the continuum of care.

According to women who participated in this study, this has a reverberation effect that deters mothers from attending facility birth. This theme is linked to the poor quality of care provided as well as distrust in the competence of providers. Previous studies in Ethiopia [42, 43] and elsewhere [44, 45] demonstrate that women experienced various forms of mistreatment in health facilities. The Ethiopian health system recognizes compassionate, patient-centered care as a priority in the efforts to improve quality and equity in service delivery, as noted in its Health Sector Transformation Plan [32]. However, much effort has not been made in improving the traditional client-provider relationships where the clinician has a paternal role to ensure women's autonomy and women-centered care. As such, it is important to give attention to strategies that address the mistreatment of childbirth regarding health workers' polite communication with clients and preserving the privacy of women as well as improving health workers' work environment.

This finding highlights that lack of trust in the obstetric skills of health center level providers is another major factor that deters women from utilizing maternal health services continually. Previous studies in Cambodia [46] and Ghana [44] also documented that some women did not have confidence in the abilities of skilled birth attendants at health center level which usually influences birthplace decision-making of mothers [44, 45]. The National Emergency Obstetric and Newborn Care assessment reports low knowledge score of health care providers regarding obstetric complications [47]. As such, this calls for prompt action to enhance obstetric skills of providers at the health center level and facilitate community-facility interface meetings to building the trust of women and communities in the health system.

Despite the expansion of primary health care, provision of free maternal health services, and free ambulance transport service from home to facilities (to ensure universal access to primary health care [32]), inaccessibility of health care service due to distance and lack of transportation are still persistent barriers for seeking maternal health care services in Ethiopia. High opportunity costs are associated with poor health service and lack of transportation which are major barriers to access maternal health services. Therefore, it is important to address these barriers to address inequalities and achieve universal access to basic health services through financial protection. This finding is confirmed by previous studies from developing countries, which demonstrate that communities with low household wealth were more unlikely to use health care services [19, 48]. Remoteness from health facilities increased community members' out-of-pocket expenditure for transportation, cost for transportation to return home, medical and food expenses which create constraints and discourage the uptake of services [48, 49]. There is still the need to further expand reach to women for skilled delivery through facility expansion, community-based outreach service delivery, or voucher scheme and health insurance program [50]. In this regard, a concerted multi-sectoral approach is needed, including improved road access, transport, and availability of physical health infrastructure.

Lack of understanding among community members, specifically mothers, regarding the availability of PNC services as well as importance of the care, is another major reason for discontinuation of uptake of service at the postpartum stage. Participants perceived PNC as necessary only if obstetric complications occur and claimed that they are not informed to return for PNC services after childbirth. Similarly, lack of understanding among community members concerning the importance of maternal health care services is reported elsewhere [48]. This shows that maternal health services in Ethiopia are fragmented where mothers' antepartum and intrapartum contacts with the health system are a missed opportunity and a significant missing link in the continuum of care [11]. The postpartum care service delivery model in Ethiopia mainly relies on home-based care provided by HEWs. Currently, women and babies are not receiving appropriate CoC at the postnatal stage in Ethiopia. The service delivery varies significantly from place to place, that is, from just discharging the woman within few hours of

delivery and letting her visit the facility only if she gets ill to having a home-visit by HEWs. Health facilities lack of space to accommodate postpartum women and preference of the family to go home to celebrate birth and taken care of other children at home are some of the reasons mentioned by respondents for early discharge. As such, there is a need to strengthen the facility-based PNC, staying for at least 24-hours after delivery and providing care as recommended by WHO and the national guide [51, 52]. Besides, a mixed-method of service delivery modality that is both facility-based and home-based PNC care, home visits by community workers, or health care providers, would be helpful for Ethiopia that has about half of an estimated 3 million annual birth cohorts happening at home [40].

This study explored the reasons for discontinuation of maternal health services across the continuum, for an in-depth understanding of the phenomena and to give a better picture to help fully understand the contextual factors in Ethiopia. Thus, the findings of this study will have significant implications on the role players to prioritize intervention approaches and build strategies to improve the utilization of maternal health services across the continuum. However, this study was subject to certain limitations. First, the findings would be prone to social desirability and recall biases since studies are cross-sectional and data are collected from self-reported recall of behavior. Another limitation would be the qualitative findings are subjective and confounded by the individual's prevailing contexts, making comparison and generalization difficult. To ensure the trustworthiness of the data, data triangulation was made, data was collected from different sources—recently delivered women, community volunteers, and community and religious leaders. Furthermore, the current study utilized IDIs and FGDs discussions to gather data, about the same phenomenon being investigated, as methodological triangulation. To ensure the credibility of the data, we framed the open-ended questions to be simple, not leading, and objective to minimize bias throughout the entire research process. The principal investigator and two research assistants who had experiences in qualitative research collected the data and we maintained neutrality throughout the data collection to not influence the participants' responses. We probed and paraphrased the participants to elaborate on or clarify what they have said during the interviews or group discussions; we also discussed the findings and interpretations of the data to avoid bias and misinterpretation of the data. Moreover, data collection approaches, notes of any decisions made on the field, raw data, analysis notes, and interpretation of data were well documented to ensure confirmability of the data.

## Conclusion

Ethiopian rural mothers still face multiple challenges to continuously uptake maternal health services. The primary maternal healthcare delivery system does not address women's needs due to poor quality of care, incompetent and unfriendly providers, disrespectful and abusive care, difficulties of transportation, and weak and timely referral links. As such, the health system has to address these problems that remain fundamental to many women to continue across the pathways in Ethiopia. Moreover, postpartum care is weakly implemented. Thus, there is an urgent need to strengthen the quality of PNC provided in facilities by implementing the national directives stating 24-hours facility stay after birth [51] and educate communities about the need to have postpartum care.

## Supporting information

**S1 Appendix. In-depth interview guide.** This is an in-depth interview guide we used to interview the participants in our study.
(DOCX)

**S2 Appendix. Focus group discussion guide.** This is a focus group discussion guide we used to elicit discussion with respondents.
(DOCX)

**S3 Appendix. IDI transcripts.** This file contains in-depth interview transcripts used for this analysis.
(DOC)

**S4 Appendix. FGD transcripts.** These are focus group discussion transcripts used for this analysis.
(DOCX)

## Acknowledgments

Addis Continental Institute of Public Health and the University of Gondar are acknowledged for providing technical support during analysis and write-up. We take this opportunity to extend our gratitude to all the study participants for the time they gave to respond to the survey questionnaires and provide us with valuable information. We would also like to acknowledge Chalachew Bekele, Ayizohbel Adamu, and Thomas Solomon for their help in data collection and transcription. Adey Abebe is acknowledged for editing the manuscript.

## Author Contributions

**Conceptualization:** Gizachew Tadele Tiruneh, Meaza Demissie, Alemayehu Worku, Yemane Berhane.

**Data curation:** Gizachew Tadele Tiruneh, Meaza Demissie, Alemayehu Worku, Yemane Berhane.

**Formal analysis:** Gizachew Tadele Tiruneh, Meaza Demissie, Alemayehu Worku, Yemane Berhane.

**Investigation:** Gizachew Tadele Tiruneh, Meaza Demissie, Alemayehu Worku, Yemane Berhane.

**Methodology:** Gizachew Tadele Tiruneh, Meaza Demissie, Alemayehu Worku, Yemane Berhane.

**Supervision:** Meaza Demissie, Alemayehu Worku, Yemane Berhane.

**Validation:** Gizachew Tadele Tiruneh, Meaza Demissie, Alemayehu Worku, Yemane Berhane.

**Visualization:** Gizachew Tadele Tiruneh, Meaza Demissie, Alemayehu Worku, Yemane Berhane.

**Writing – original draft:** Gizachew Tadele Tiruneh.

**Writing – review & editing:** Gizachew Tadele Tiruneh, Meaza Demissie, Alemayehu Worku, Yemane Berhane.

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
