## [Decision Letter · Decision Letter 0]

19 May 2021

PONE-D-20-38899

Community’s experience and perceptions of maternal health services across the continuum of care in Ethiopia: a qualitative study

PLOS ONE

Dear Dr. Tiruneh,

Thank you for submitting your manuscript to PLOS ONE. After careful consideration, we feel that it has merit but does not fully meet PLOS ONE’s publication criteria as it currently stands. Therefore, we invite you to submit a revised version of the manuscript that addresses the points raised during the review process.

Your manuscript has undergone the peer-review process and the reviewers have provided their comments/suggestions. Kindly address these points/concerns before we make a decision.

We look forward to receiving your revised manuscript.

Kind regards,

Kingston Rajiah

Academic Editor

PLOS ONE

Journal Requirements:

[Addis Continental Institute of Public Health (ACIPH) and the University of Gondar (UoG) are acknowledged for providing support during analysis and write up.]

 [The author(s) received no specific funding for this work.]

4. Thank you for stating the following in the Financial Disclosure section:

[The author(s) received no specific funding for this work.].   

We note that one or more of the authors are employed by a commercial company: JSI Research & Training Institute, Inc.

Reviewers' comments:

Reviewer's Responses to Questions

**Comments to the Author**

1. Is the manuscript technically sound, and do the data support the conclusions?

Reviewer #1: Yes

Reviewer #2: Partly

Reviewer #3: Partly

2. Has the statistical analysis been performed appropriately and rigorously? 

Reviewer #1: N/A

Reviewer #2: N/A

Reviewer #3: N/A

3. Have the authors made all data underlying the findings in their manuscript fully available?

Reviewer #1: Yes

Reviewer #2: Yes

Reviewer #3: Yes

4. Is the manuscript presented in an intelligible fashion and written in standard English?

Reviewer #1: Yes

Reviewer #2: Yes

Reviewer #3: Yes

5. Review Comments to the Author

Reviewer #1: General comments

Maternal and child health is still a priority of concern in Ethiopia. I read this manuscript with pleasure and I found it an interesting topic. In general, the study is presented logically and scientifically, well-edited, and it is in a standard English language. I have only a few comments mostly on clarification.

Q1. Under the methods(L120 – 124): The sentence starting with “ where I am providing technical assistance…) is confusing and not important to indicate who you are working for. In this study, there is more than one author and it is unknown who I refer to. This could also raise an issue of conflict of interests if the project owner is also the principal investigator of this study. I suggest either modifying this paragraph or avoiding using first-person singular pronouns.

Q2: Data collection: When were the themes developed? Was it before or after data collection? It is mentioned that you have used both deductive and inductive coding, but it is not clear if themes were developed after the interview based on the findings or you come up with pre-identified themes. Please also justify the reason to choose either one or both methods.

Q3. Results:

You have mentioned that the study used participants from high and low-performing districts(L137-139). However, in your results, when you quote the participants' thoughts, you didn't indicate whether the participant is from(high performing or low-performing district). In my opinion, it would be more informative if you could add where the participant is from( high or low performing district). E.g. ……………(Community leader, High performing district, IDI).

Q4. Although this study is not comparative, I think it is possible to narrate if there is any difference between the high performing and low performing districts with respect to your study themes. Are there any best practices or lessons learned from the high-performing districts? This can be included in your result or discussion part unless I have missed it.

Q5. The phrase(L314) “fear of the growth of the baby” is not clear. What does fear of the growth of the baby mean and how it could be a reason for discontinuation of PNC services?

Reviewer #2: 1. Title: In this study, authors mainly shared the community’s experience rather than perceptions; so I will suggest to remove the word “perception” from the title, and also to make the necessary changes throughout the document including the research questions and keywords.

2. Keywords: Suggesting to remove “postnatal care” as it is the part of continuum of care. Community has negative experience not only related to the postnatal care but also to the two other aspects (ANC and intrapartum) of continuum of care.

3. Financial disclosure: Without having any specific fund, how the data collection and data analysis chapters were managed? Is this study part of the L10K project? Is there any incentive or token of appreciation for the FGD and IDI participants? If not, what’s motivate them to participate in this study?

4. In Abstract: Number of FGD and IDI needs to mention clearly.

5. In the background, the definition of continuum of care needs to explain clearly. To justify the problem statement properly, the overall maternal health situation needs to describe including the trends of MMR, and how it differs between urban and rural areas. This study’s main focus is at the primary level health care which is not reflected in the background.

6. Method section: This is a qualitative study where purposive sampling is acceptable; even then, please mention “what are the efforts the investigators made to avoid biasness”, as at line 120-121, it was mentioned “how the principal investigator is involved with the study areas”, whether this working relationship affects the study findings or not. It was mentioned that the data analysis (line 189) is guided by the conceptual framework but I did not find the framework. I am suggesting to add the brief description of the framework in the method section

7. Results: Please add the age range for all kinds of participants in addition to their mean age. Line 219-220, please explain with references- what are the evidences of utilizing ANC, facility delivery, infant vaccination and postpartum family planning services by most of the women? Is it the study findings or overall Ethiopian’ maternal health situation? Throughout the document, I did not find any positive experience, almost all are the negative experiences mentioned by study participants. If the study participants have any positive experience which authors did not add to this manuscript or if there is any evidence, better to present with reference to justify this statement.

Mothers and the community are not aware of PNC service. As there is a national guideline [Ref 42], authors should discuss about this guideline early instead of mentioning at the last line of the conclusion; also “what are the barriers to implement this guideline”- need to discuss.

Why health promotion activity did not focus on PNC services? Who is the implementer of the health promotion activity, whether government or non- government/private agency? -- need to discuss.

Line 241- What is the reason of “unknown LMP” as it leads to delayed booking?

Line 324-“did not realize she is pregnant until the 4th or 5th month”; why these women did not notice their missed menstrual period? whether these women were having irregular menstrual period or not. All these issues need to be find out by the researchers?

Line 369 -370- as this study area is located in rural area; how the respondents are comparing their experience with that of urban patients; this is not clear to me

Line-379- “women give birth without disclosing their pregnancy”- can authors explain how women hide their 9-month pregnancy period? How the community or family members help them to hide it?

Line 523_ “unduly exposure of women’s reproductive organs and lack of privacy”- what does this mean? Whether the vaginal examination is done in open space in presence of other people who are not service providers. Because, at the line 362- the community leaders said that “he was waiting outside far from the delivery room” i.e. even husband was not allowed to stay with wife.

8. Discussion: This section is relatively weak and needs to be strengthening by explaining the study findings with proper justification and evidences. In this section, authors re-emphasised the study findings with some justification which is not sufficient. The study findings need to be explained by the authors. Example:

Line 535 – “lack of trust” – what is the suggestion of authors to improve the trust?

Line 558- why community people does not know about PNC service if it is available? What are the main barriers?

Line 563- “maternal health services are weak” whether authors can make this conclusion based on this qualitative study? To justify this statement, authors need to mention other evidences if available?

Line 565—566- this is confusing; if it is the responsibility of health workers to provide PNC at home; then why we are expecting that health care providers of the health center will suggest the mothers to come at the facility for PNC?

Line 571-574: If the authors suggested for “mixed-method of PNC service”, what will be the effective implementation strategy to avoid the duplication of services between health workers and health care providers, and how to ensure the proper implementation of PNC services. This needs to be explained clearly by authors.

9. Minor issues:

a. In-text citation should be within square bracket

b. Before using any abbreviation, it needs to spell out the full term first e.g. line 63 (COC), line 84 (EDHS), line 108 (HEP), line 120 (L10K); please check all abbreviation and make necessary changes following the rules of abbreviations. The list of abbreviation is also missing

c. Birr is the unit of currency in Ethiopia; it needs to be presented at the international currency unit like USD; otherwise it will be difficult for the international readers to understand it clearly

d. In the same way, authors should explain “kebeles”;

e. Line 198- please check the sentence to complete it

f. Community leader vs community elders (line 414, 426, 585) – needs to be consistent as the operational definition of leaders and elders are different.

g. At the end of each quotation whether the anonymous identifier can be part of the sentence e.g. “……………………………………….” (recently delivered mother, FGD).

h. What is the gender of community leaders? Are they all males or mixed i.e. some are males and rest are females, please mention it?

i. Line 133- community volunteers – do they receive any incentive for their work?

Reviewer #3: Good article and informative. Minimal errors need to be addressed:

1) Some abbreviations ie. CoC and HEW were not put in a full meaning - take note for the abbreviation in the article and the rest will follow the abbreviation. Please look at the introduction section.

2) How does author ensure the adequacy of the sample size in this study? Did not mention on whether saturation has been reached or not.

3) The word National in the National Emergency Obstetric and Newborn Care was put as small letter. The letter 'N' should be in capital letter.

6. PLOS authors have the option to publish the peer review history of their article (what does this mean?). If published, this will include your full peer review and any attached files.

Reviewer #1: **Yes: **Serebe Gebrie

Reviewer #2: No

Reviewer #3: No

---

## [Author Response · Author response to Decision Letter 0]

14 Jul 2021

Point-by-point response to reviewer/editorial

Journal: PLOS ONE

Title: Community’s experience and perceptions of maternal health services across the continuum of care in Ethiopia: a qualitative study (PONE-D-20-38899)

The authors would like to appreciate and thank the reviewers for the constructive comments.

Our point-by-point responses to the reviewers are below each of the comments in italics. We also make sure that this version of the manuscript conforms the journal style

Journal Requirements:

Updated funding and conflict of interest statements

Funding declaration: The authors declare that they did not receive funding for this research from any source. JSI Research & Training Institute, Inc. has provided us support in the form of salaries for author [GT]. However, any of the funders did not have role in study design, data collection and analysis, decision to publish, or preparation of the manuscript.

Competing interest: The authors declare that they have no competing interests. One of the authors have been working for JSI Research & Training Institute, Inc., a commercial company. We declared that this commercial affiliation does not alter our adherence to PLOS ONE policies on sharing data and materials.

Reviewers' comments:

Reviewer #1: General comments

Maternal and child health is still a priority of concern in Ethiopia. I read this manuscript with pleasure and I found it an interesting topic. In general, the study is presented logically and scientifically, well-edited, and it is in a standard English language. I have only a few comments mostly on clarification.

Q1. Under the methods (L120 – 124): The sentence starting with “ where I am providing technical assistance…) is confusing and not important to indicate who you are working for. In this study, there is more than one author and it is unknown who I refer to. This could also raise an issue of conflict of interests if the project owner is also the principal investigator of this study. I suggest either modifying this paragraph or avoiding using first-person singular pronouns.

Comment well taken and addressed in this version.

Q2: Data collection: When were the themes developed? Was it before or after data collection? It is mentioned that you have used both deductive and inductive coding, but it is not clear if themes were developed after the interview based on the findings or you come up with pre-identified themes. Please also justify the reason to choose either one or both methods.

Thank you for the comments. First, we identified codes/constructs based on the theoretical framework and the IDI and FGD guides. Then, through reading and rereading of the transcripts and listening of the audios, we also produced new codes. Then we categorize and thematize the concepts. 

Q3. Results:

You have mentioned that the study used participants from high and low-performing districts(L137-139). However, in your results, when you quote the participants' thoughts, you didn't indicate whether the participant is from(high performing or low-performing district). In my opinion, it would be more informative if you could add where the participant is from( high or low performing district). E.g. ……………(Community leader, High performing district, IDI).

Q4. Although this study is not comparative, I think it is possible to narrate if there is any difference between the high performing and low performing districts with respect to your study themes. Are there any best practices or lessons learned from the high-performing districts? This can be included in your result or discussion part unless I have missed it.

Comment well noted. We recruited three groups of respondents from two different contexts, better performing and low performing districts, to yield a wider perspective from various groups of stakeholders and contexts Accordingly, we have edited the sampling methods employed into maximum variation sampling schemes 

Q5. The phrase(L314) “fear of the growth of the baby” is not clear. What does fear of the growth of the baby mean and how it could be a reason for discontinuation of PNC services?

Thank you for picking this error. It is to mean women’s perceived fear of overgrowth of the fetus if pregnant women attended frequent ANC visits during pregnancy. For that perception and fear, they discontinue for subsequent and more ANC visits. It is now corrected. 

Reviewer #2: 1. Title: In this study, authors mainly shared the community’s experience rather than perceptions; so I will suggest to remove the word “perception” from the title, and also to make the necessary changes throughout the document including the research questions and keywords.

Thank you so much for the valid comments. The communities’ perceptions regarding the need for maternal and newborn health services, perceived reasons for recipient of recommended services, and discontinuation across the continuum of care is now elaborated, besides their experiences. 

2. Keywords: Suggesting to remove “postnatal care” as it is the part of continuum of care. Community has negative experience not only related to the postnatal care but also to the two other aspects (ANC and intrapartum) of continuum of care.

comment well taken. 

3. Financial disclosure: Without having any specific fund, how the data collection and data analysis chapters were managed? Is this study part of the L10K project? Is there any incentive or token of appreciation for the FGD and IDI participants? If not, what’s motivate them to participate in this study?

Thanks for the comments. The principal investigator covers the cost involved for data collection. Study participants were not paid for their participation in the study. We took their consent of participation. Besides, we conducted the data collection during the weekends and holly days. 

4. In Abstract: Number of FGD and IDI needs to mention clearly.

comment well taken and addressed in the revised version 

5. In the background, the definition of continuum of care needs to explain clearly. To justify the problem statement properly, the overall maternal health situation needs to describe including the trends of MMR, and how it differs between urban and rural areas. This study’s main focus is at the primary level health care which is not reflected in the background.

Comment well taken. The following paragraph is added. “Maternal and newborn health is a major public health problem in low-income countries. Sub-Saharan Africa (SSA) is the only region with unacceptably high maternal mortality ratio, accounted for about two-thirds of global maternal deaths in 2017 [1]. Though Ethiopia achieved a substantial reduction in maternal death of roughly 61% between 2000-2017 [1], its maternal mortalities are still among the highest in the world [1-3] and persistently high neonatal mortality since the last two decades [4-6]. The utilization of maternal and newborn healthcare services across the continuum remains low in SSA [7-9]. The uptake of services drastically declined from antenatal to the postnatal period, along with the CoC [10] where coverage is lowest during childbirth and postnatal period, and services are often fragmented and weakly implemented, limiting continuity of care [11]. “

6. Method section: This is a qualitative study where purposive sampling is acceptable; even then, please mention “what are the efforts the investigators made to avoid biasness”, as at line 120-121, it was mentioned “how the principal investigator is involved with the study areas”, whether this working relationship affects the study findings or not. 

Sure, the principal investigator has prolonged engagement in the area during data collection with the study participants undertaking interviews and focus group discussions as well as well the principal investigator have been engaged long in the study area during the project implementation. This would help to well understand the community norms, beliefs, and language and get adequate information. To minimize biases, we undertook different methods including framing open-ended questions to prevent the participant from simply agreeing or disagreeing, and guide participants to provide a truthful and honest answer and engage them throughout the interview and maintain neutrality so as to not influence the participants’ responses. These are elaborated in the manuscript.

It was mentioned that the data analysis (line 189) is guided by the conceptual framework but I did not find the framework. I am suggesting to add the brief description of the framework in the method section

Comment well taken. The framework is now elaborated. Line 127-135

7. Results: Please add the age range for all kinds of participants in addition to their mean age. 

Comment well taken and addressed. 

Line 219-220, please explain with references- what are the evidences of utilizing ANC, facility delivery, infant vaccination and postpartum family planning services by most of the women? Is it the study findings or overall Ethiopian’ maternal health situation? Throughout the document, I did not find any positive experience, almost all are the negative experiences mentioned by study participants. If the study participants have any positive experience which authors did not add to this manuscript or if there is any evidence, better to present with reference to justify this statement.

Evidence show that there is increasing trend in the uptake of ANC, facility delivery, infant vaccination and postpartum family planning services in Ethiopia. Over the last decades, institutional delivery increased 5-fold and most mothers and more than three-fourth of mothers utilizing ANC, according to EDHS. This has elaborated in the revised version of the manuscript. Line 501-504 

Mothers and the community are not aware of PNC service. As there is a national guideline [Ref 42], authors should discuss about this guideline early instead of mentioning at the last line of the conclusion; also “what are the barriers to implement this guideline”- need to discuss.

Why health promotion activity did not focus on PNC services? Who is the implementer of the health promotion activity, whether government or non- government/private agency? -- need to discuss.

Thanks a lot for the valid comments: According the national guideline, postpartum home visit is a responsibility of HEWs, frontline community health workers not health care providers. Besides, mothers are expected to visit health facilities for PNC. Reasons for early discharge are discussed in the revised version. 

Line 241- What is the reason of “unknown LMP” as it leads to delayed booking?

Line 324-“did not realize she is pregnant until the 4th or 5th month”; why these women did not notice their missed menstrual period? whether these women were having irregular menstrual period or not. All these issues need to be find out by the researchers?

Respondents mentioned that due to use of contraceptives, there were irregularities in menstrual cycles. The following sentence is added to the results to elaborate more. “Moreover, according to the accounts of respondents, family planning methods women used might cause an absence of menstrual period and this might mislead women, becoming a reason for them not to go and test for pregnancy and thus the reason for not start their ANC early.” Line 251-53

Line 369 -370- as this study area is located in rural area; how the respondents are comparing their experience with that of urban patients; this is not clear to me

Sure, through respondents are from rural settings, those who are catchments of the woreda town health centers came to visit the town health center. 

Line-379- “women give birth without disclosing their pregnancy”- can authors explain how women hide their 9-month pregnancy period? How the community or family members help them to hide it?

It means to say, they don’t want to disclose their pregnancy to their relatives or people in their community/village that are far from them. As the settlement of the rural community is very sparse, they could hide themselves not disclosing to all their community and relatives far away from them. They could also avoid social events or gatherings like market places, churches, mosques, etc. Those who are near to their house would definitely know. 

Line 523_ “unduly exposure of women’s reproductive organs and lack of privacy”- what does this mean? Whether the vaginal examination is done in open space in presence of other people who are not service providers. Because, at the line 362- the community leaders said that “he was waiting outside far from the delivery room” i.e. even husband was not allowed to stay with wife.

It means that naked and lithotomy position exposes their reproductive organ to those assistant health care providers, apprenticeship students, and other laboring women not only during vaginal examination during the whole laboring process. 

8. Discussion: This section is relatively weak and needs to be strengthening by explaining the study findings with proper justification and evidences. In this section, authors re-emphasised the study findings with some justification which is not sufficient. The study findings need to be explained by the authors. Example:

Line 535 – “lack of trust” – what is the suggestion of authors to improve the trust?

comment well taken and addressed in the revised version. Line 565-6

Line 558- why community people does not know about PNC service if it is available? What are the main barriers?

Postnatal care is weakly implemented in the country. It is less prioritized and not integrated with ANC and immunization services well. Accordingly, health care providers are not providing adequate counseling services to mothers regarding importance of PNC and place of PNC. This is well articulated in the manuscript. 

Line 563- “maternal health services are weak” whether authors can make this conclusion based on this qualitative study? To justify this statement, authors need to mention other evidences if available?

Comment well taken and addressed in the revised version

Line 565—566- this is confusing; if it is the responsibility of health workers to provide PNC at home; then why we are expecting that health care providers of the health center will suggest the mothers to come at the facility for PNC? Line 571-574: If the authors suggested for “mixed-method of PNC service”, what will be the effective implementation strategy to avoid the duplication of services between health workers and health care providers, and how to ensure the proper implementation of PNC services. This needs to be explained clearly by authors.

Thanks a lot for the valid comments: According the national guideline, postpartum home visit is a responsibility of HEWs, frontline community health workers not health care providers. Besides, mothers are expected to visit health facilities for PNC. Reasons for early discharge are also discussed in the revised version. 

9. Minor issues:

a. In-text citation should be within square bracket

Well noted. It is now changed into SpringerVancouverNumber that is with a square bracket as recommended by the Journal.

b. Before using any abbreviation, it needs to spell out the full term first e.g. line 63 (COC), line 84 (EDHS), line 108 (HEP), line 120 (L10K); please check all abbreviation and make necessary changes following the rules of abbreviations. The list of abbreviation is also missing

Thanks much. All acronyms are spelled out in the text and list of acronyms are now presented (Line 608)

c. Birr is the unit of currency in Ethiopia; it needs to be presented at the international currency unit like USD; otherwise it will be difficult for the international readers to understand it clearly

Thanks. Comment well taken and addressed in this version. 

d. In the same way, authors should explain “kebeles”;

comment well taken and defined the revised version as Kebele is the lowest administrative units of the country. 

e. Line 198- please check the sentence to complete it

edited; thanks

f. Community leader vs community elders (line 414, 426, 585) – needs to be consistent as the operational definition of leaders and elders are different.

Thanks; well addressed

g. At the end of each quotation whether the anonymous identifier can be part of the sentence e.g. “……………………………………….” (recently delivered mother, FGD).

Addressed 

h. What is the gender of community leaders? Are they all males or mixed i.e. some are males and rest are females, please mention it?

All were males. And mentioned in the methods section 

i. Line 133- community volunteers – do they receive any incentive for their work?

Though there is no uniform and standard performance-based incentive mechanisms. There are some form of incentives including non-financial (like recognition) and provision of in kind items like uniforms, shoes, etc. 

Reviewer #3: Good article and informative. Minimal errors need to be addressed:

1) Some abbreviations ie. CoC and HEW were not put in a full meaning - take note for the abbreviation in the article and the rest will follow the abbreviation. Please look at the introduction section.

Thanks a lot for the comments. We addressed these and similar errors. 

2) How does author ensure the adequacy of the sample size in this study? Did not mention on whether saturation has been reached or not.

Thanks a lot for the valuable comments. The following sentence is added in this version. “The principal investigator and research assistants collected and analyzed informational redundancy, data saturation, after conducting three FGDs and six IDIs (30).” Line 145-6

3) The word National in the National Emergency Obstetric and Newborn Care was put as small letter. The letter 'N' should be in capital letter.

Thanks again. It is addressed.

---

## [Editor Report · Decision Letter 1]

16 Jul 2021

Community’s experience and perceptions of maternal health services across the continuum of care in Ethiopia: a qualitative study

PONE-D-20-38899R1

Dear Dr. Tiruneh,

We’re pleased to inform you that your manuscript has been judged scientifically suitable for publication and will be formally accepted for publication once it meets all outstanding technical requirements.

Kind regards,

Kingston Rajiah

Academic Editor

PLOS ONE
---

## [Editor Report · Acceptance letter]

22 Jul 2021

PONE-D-20-38899R1 

Community’s experience and perceptions of maternal health services across the continuum of care in Ethiopia: a qualitative study 

Dear Dr. Tiruneh:

I'm pleased to inform you that your manuscript has been deemed suitable for publication in PLOS ONE. Congratulations! Your manuscript is now with our production department. 

Kind regards, 

on behalf of

Dr. Kingston Rajiah 

Academic Editor

PLOS ONE